# Copepod Community Structure in Pre- and Post-Winter Conditions in the Southern Adriatic Sea (NE Mediterranean)

**Marijana Hure [1],\*** , **Mirna Batistić [1]**, **Vedrana Kovačević [2]** , **Manuel Bensi [2]** and **Rade Garić [1]**

[1]  Institute for Marine and Coastal Research, University of Dubrovnik, Kneza Damjana Jude 12, 20000 Dubrovnik, Croatia; mirna.batistic@unidu.hr (M.B.); rade.garic@unidu.hr (R.G.)

[2]  National Institute of Oceanography and Applied Geophysics—OGS, 34010 Sgonico (TS), Italy; vkovacevic@inogs.it (V.K.); mbensi@inogs.it (M.B.)

\*  Correspondence: marijana.hure@unidu.hr; Tel.: +385-20323515

**Abstract:** Copepod communities were studied along an east-west transect in the oligotrophic Southern Adriatic Sea. This dynamic region is under the influence of various physical forces, including winter vertical convection, lateral exchanges between coastal and open sea waters, and ingression of water masses of different properties all of which occurred during the investigation periods. Depth-stratified samples were taken with a Nansen net (250 μm mesh size) in pre- and post-winter conditions in 2015/2016. In December, the coastal copepod community was limited over the western flank, while epipelagic waters of the open and eastern waters were characterized by high diversity, low abundances in the central area, and subsurface/upper mesopelagic copepod species. In April, higher abundances were recorded over the entire vertical profile with the surface coastal copepod community present through the entire transect. Higher abundances in the central area during the post-winter period are probably a consequence of late-winter/early spring blooms near the center of the Southern Adriatic. Mesopelagic fauna of both months was characterized by high abundances of *Haloptilus longicornis*, characteristic species of the eastern Mediterranean, whose larger presence was favored by the cyclonic phase of the North Ionian Gyre and a consequent strong Levantine Intermediate Water ingression.

**Keywords:** mesozooplankton; spatial distribution; copepods; community composition; water masses distribution; Adriatic Sea; Eastern Mediterranean

## 1. Introduction

The Adriatic Sea is an elongated and relatively shallow basin, stretching south-eastward for 800 km from the highest Mediterranean latitude (45°47′ N) in the Eastern Mediterranean basin.

The current regime of the whole Adriatic Sea is characterized by a basin wide cyclonic surface flow: On the eastern flank, the so called Eastern Adriatic Current (EAC) flows northward and brings warm and salty waters to the northernmost part of the Adriatic. On the western flank, instead, the Western Adriatic Current (WAC) flows south-eastward geostrophycally balanced along the Italian coast, transporting relatively cold and fresh water out of the Adriatic Sea [1–5], through the Strait of Otranto (~90 km wide and ~800 m deep), which is the communication channel between the Adriatic and the Ionian seas. The major freshwater component of the WAC is provided by the Po River outflow, which has a mean daily river run-off of about 1500 m³/s and maximum peaks of 10,000 m³/s [6,7].

The Southern Adriatic (SA, Figure 1) sub-basin represents the deepest part of the Adriatic, with a maximum depth of ~1200 m. It is also characterized by the presence of a quasi-permanent cyclonic South Adriatic Gyre [3,8,9]. The hydrodynamics of the SA is influenced by occasionally very strong

episodes of cold and dry northerly winds and warm and wet southerly winds as well as by a regular exchange of water with the adjacent Ionian Sea through the Strait of Otranto and with the northern part of the Adriatic Sea. Furthermore, the SA physical and biogeochemical water properties are strongly influenced by local dense water formation events and by the North Ionian Gyre (NIG) multiannual reversals that are regulated mainly by the Adriatic-Ionian Bimodal Oscillation system (BiOS) [10,11]. The BiOS regulating system determines the alternating prevalent entrance into the Adriatic Sea of Levantine Intermediate Waters (LIW, cyclonic NIG) or of Atlantic Water (AW, anti-cyclonic NIG) from the Ionian Sea. This process is able to regulate the salt and nutrients content of the entire basin [10].

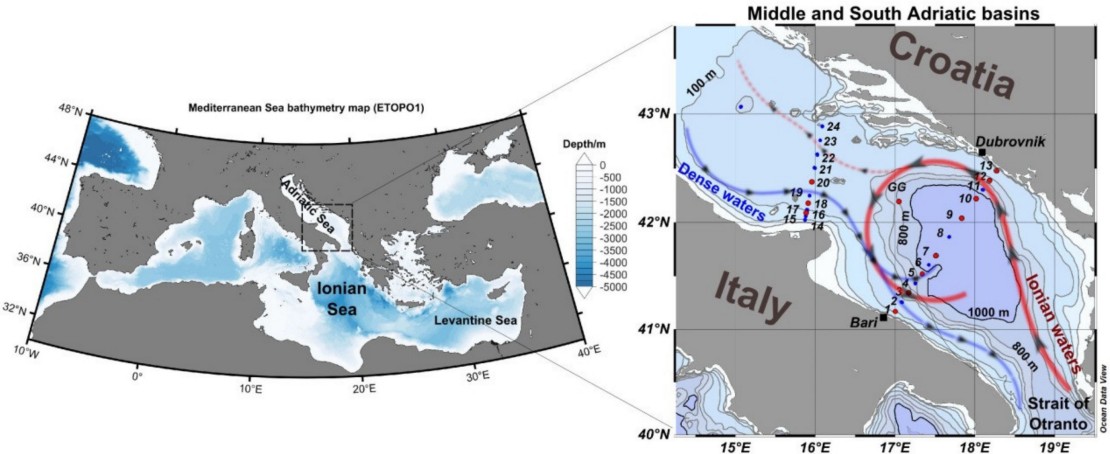

**Figure 1.** Bathymetric map of the Mediterranean Sea with a zoom (right panel) of the study area in the Adriatic Sea. Conductivity-temperature-depth (CTD) stations (blue dots) of the Evolution and Spreading of the Southern Adriatic Waters (ESAW) cruises in December 2015 (10th–15th) and in April 2016 (5th–10th). Superimposed red dots indicate zooplankton sampling stations. Stations 14–24 form the Middle Adriatic Transect, and stations 1 to 14 form the South Adriatic Transect. Blue and red lines show a schematic of the south Adriatic winter circulation.

The exchanges of waters with the Ionian Sea and the occurrence of strong winter vertical mixing of the water column have also large impacts on the occurrence of phytoplankton blooms and hence on the primary production in the Adriatic Sea [12–14] as well as on the abundance and distribution of marine species [14–20].

Copepods, as a dominant component of marine zooplankton in terms of biomass and diversity [21–26] influence numerous aspects of ecosystem function. They are the main trophic link between the basis of the food web and the upper trophic levels [27–29]. Moreover, through vertical migration they actively participate in the transport of particles to the deeper ocean [30] and produce rapidly sinking faecal pellets after grazing on primary producers in the euphotic layer [31]. Apart from the differentiation of copepod community composition and structure with depth [32], they are also affected by primary production and hydrographic conditions [33,34], which influence food availability and species depth range.

Detailed investigations of the copepods in the SA began in the middle of last century with studies on their annual production cycles, horizontal and vertical distributions, and diel vertical migration patterns [35–43]. The SA is inhabited by almost the entire Adriatic copepod fauna [44]. In contrast to the northern part, it is characterized by notable species diversity and the presence of a large number of oceanic species that comprise relatively uniform but quantitatively poor populations [44,45]. Hure et al. [45] distinguished the oceanic copepod community of the SA in three depth distribution zones: Upper zone species, middle zone species, and the lower zone. They also described their seasonal vertical distribution. Miloslavić et al. [46], instead, gave information on the composition, numerical abundance, and vertical structure of micro and mesozooplankton across the coastal and offshore waters

of Albania. Batistić et al. [15] described the zooplankton community, including copepods, by analyzing data collected during the winter convection event recorded in the deep SA in February 2008. Finally, the winter community structure of the mesozooplankton related to water-masses in the eastern SA was described by Hure et al. [19]. However, a recent detailed review focused on copepod community composition over the SA is lacking.

Therefore, the aim of this study is to present copepod assemblages with respect to different environmental conditions in pre- and post-winter periods over the SA. Special attention is paid to the influence of winter vertical convection, lateral exchanges between coastal and open sea waters, and the ingression of water masses of different properties into the Adriatic on copepod abundance and distribution. In addition, we test the hypothesis that the number of copepods increases significantly at the open oligotrophic SA in the post-winter vertical convection period and that their abundance is equal to or greater than in more eutrophic coastal area.

## 2. Materials and Methods

The oceanographic and biological data were collected during the two research cruises, named ESAW (Evolution and Spreading of the Southern Adriatic Waters) conducted in the Middle and Southern Adriatic, in December 2015 (from the 10th to 15th) and in April 2016 (from 5th to 10th) (Figure 1) in the framework of the Eurofleets2 programme. Vertical profiles of temperature, salinity, chlorophyll-*a* fluorescence (Chl-*a*), and dissolved oxygen (DO) concentration (averaged over 1-dbar intervals) were taken by means of CTD (Conductivity-Temperature-Depth) multiparametric probe SBE 911 plus (SEA Bird Electronics INC., USA) equipped with a WETLabs Fluorometer and SBE43 oxygen sensors. Derived parameters, such as the potential temperature (θ) and potential density anomaly referred to 0 dbar were calculated from original in situ data using the TEOS-10 Gibbs function approach (IOC, SCOR, and IAPSO, 2010). Hereafter we will use θ to refer to the seawater temperature.

Zooplankton samples from selected stations (Figure 1) are presented in Table 1. In addition to the SA transect, three zooplankton sampling stations in the Middle Adriatic were included in this investigation for better insight into the influence of less saline northern Adriatic waters on the copepod distribution. Zooplankton sampling was done by vertical tows using a Nansen opening-closing net with 250 μm mesh (113 cm diameter; 380 cm length). In total, 68 samples were collected, depending on station depth, within the layers 0–50, 50–100, 100–200, 200–300, 300–400, 400–600, 600–800, and 800–1200 m. The average hauling speed of the tows was 0.5–1 m/s. On board, samples were fixed and preserved in a seawater-formalin solution containing about 2.5% formaldehyde buffered with $CaCO_3$. The counting and taxonomic identification of the copepods were performed under an Olympus SZX16 stereomicroscope on subsamples ranging from 1/7 to 1/10 of each sample, depending on the total sample abundance. Each entire sample was examined for the identification of rare species. Taxonomic identification was performed to a species level for the majority of the adults. Some groups were identified at higher taxonomic levels (e.g., copepodite stages, *Oithona setigera*-group, which included individuals of *O. setigera*, *O. longispina* and *O. atlantica*, genera *Copilia*, *Vettoria* and *Sapphirina* and families Corycaeidae and Oncaeidae). Abundance is expressed as individuals per cubic metre (ind. m$^{-3}$).

For univariate biodiversity measures, Margalef's species richness (d) and Shannon-Wiener diversity index (H′) were calculated for each sample using PRIMER 5 for Windows software [47]. The Margalef richness formula [48] compares the number of taxa in a sample and the total number of organisms comprising those taxa. The Margalef's species richness is given by the equation: d = (S − 1)/ln × N, where S is the number of taxa in the sample and N is the total number of individuals. The Shannon-Wiener index [49] evaluates how the individuals are distributed among the taxa and is determined by the equation: H′ = −Σi × Pi × lnPi, where Pi is the proportion that the i-th species represents to the total number of individuals in the sampling space.

Patterns in the community structure were determined by using PC-ORD for Windows 5.10 [50]. All data were log10(N + 1) transformed. Cluster analysis was used to identify natural groupings at sampling stations and depths based on similarity in the copepod community structure; similarities

between samples with abundances for taxa were analyzed by calculating the relative Euclidean distance measure, using Ward's linkage method [51]. The Euclidean coefficient is generally used as a distance measure for computing sample resemblances when the data consist of quantitative abundance data [52]. The resulting matrix was used to generate a dendrogram scaled by both the objective function and by percentage of information remaining. Groups were identified by pruning to 45% retained information.

**Table 1.** List of samples collected in the southern Adriatic in December 2015 and April 2016.

| | Stations | | | | | | | | | | | | |
|---|---|---|---|---|---|---|---|---|---|---|---|---|---|
| **Depth layers** | **1** | **3** | **5** | **7** | **9** | **10** | **12** | **13** | **16** | **18** | **20** | **GG** |
| 0–50 | x | | x | | x | x ** | x | x | | | | x |
| 50–100 | x | | x | | x | x ** | x | x | | | | x |
| 0–100 | | x | | x * | | | | | x * | x | x | |
| 100–200 | | x ** | x | | x | x ** | x | | | | | x |
| 200–300 | | | x | | x | x ** | x | | | | | x |
| 300–400 | | | x * | | x | x ** | | | | | | x |
| 400–600 | | | x * | | x | x ** | | | | | | |
| 600–800 | | | x * | | x * | x ** | | | | | | |
| 800–1200 | | | | | x * | x ** | | | | | | |

\* Sampled only in December; ** sampled only in April.

Afterwards, the Indicator Species Analysis (ISA; [53]) was used to identify taxa representative of the different clusters. This method combines information on the concentration of species abundance in a particular group and the consistency of occurrence of a species in a particular group. It produces indicator values (IV) for each species in each group with a range from zero (no indication) to 100 (perfect indication). Only taxa with IV > 25% were considered characteristic of the clusters [53]. Finally, a Monte Carlo randomization was used to assess the statistical significance of the IVs (alpha = 0.001).

In order to detect relationships between zooplankton communities (clusters) and environmental variables (sampled layer, season, temperature, salinity, Chl-*a*, and DO) a Non-Metric Multidimensional Scaling (NMS) was used. The NMS ordination was performed using the Sorensen (Bray-Curtis) distance measure. Dimensionality was determined by evaluating the standard residual sum of squares (STRESS; [54]). STRESS values lower than 20 indicate a stabile solution [51]. Environmental variables were correlated to ordination axes using Pearson's r.

## 3. Results

### 3.1. Environmental Conditions

Data acquired during the two oceanographic cruises characterized two different conditions typically associated with the late-autumn and early spring seasons. In the SA, the surface layer was slightly warmer and fresher (14.56 < $\theta$ < 18.13 °C, 36.98 < S < 38.83) in December 2015 than in April 2016 (13.86 °C < $\theta$ < 17.17 °C, 37.42 < S < 38.94). The winter phase was characterized by a weak vertical convection that reached not more than 400 m. The minimum temperature value of 13.09 °C was recorded in the deepest part of the water column at the station of ESAW-9 both in December and in April (Figure 2a,b). The salinity in the upper layer was lower near the shore than in the open sea in both sampling periods, with a pronounced minimum (36.98/37.42) found on the western flank (Figure 2d,e). The inflow of saltier water from the Ionian Sea (i.e., LIW) along the eastern flank of the SA was favoured by the still active cyclonic phase of the NIG circulation. The greatest salinity values (38.95) were recorded in December 2015 at station ESAW-11 between 160 and 170 m depth and in April (38.94) at station ESAW-10 at 20–60 m depth.

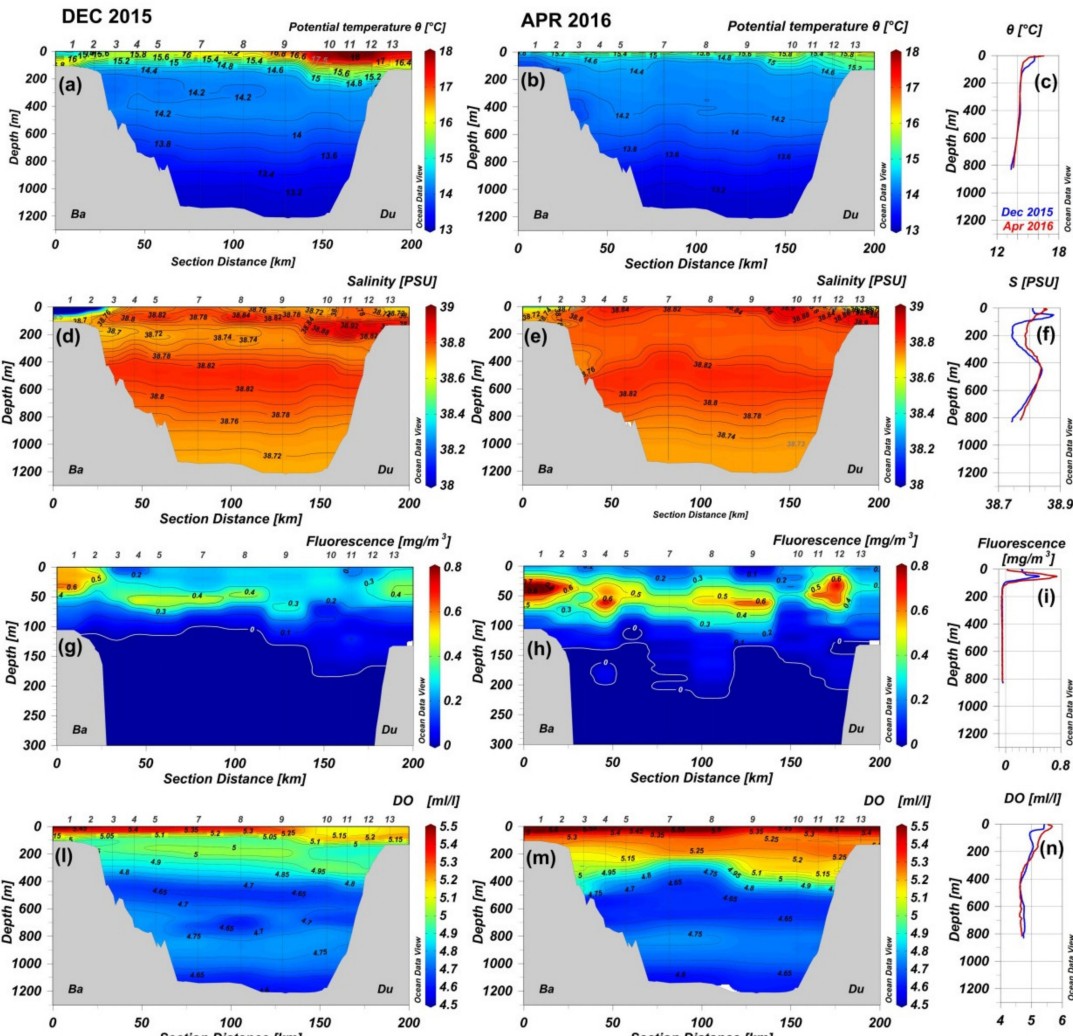

**Figure 2.** Vertical distribution of potential temperature (**a–c**), salinity (**d–f**), chlorophyll *a* (by fluorescence), (**g–i**) and dissolved oxygen concentration (**l–n**) along the investigated transect in December 2015 and in April 2016 (from Njire et al., 2019 [20]). For the northernmost station ESAW-GG (see Figure 1 for its location) the corresponding vertical profiles are presented separately.

Furthermore, data revealed the presence of a double salinity maximum, unusual for the SA [55] and more noticeable in December. The first salinity maximum was located roughly between 50 and 200 m depth, and the other between 400 and 600 m depth with relatively fresh water (38.72–38.74) in between. Generally, subsurface fluorescence concentrations decreased from the western toward the eastern side of the SA (Figure 2g,h). In April, fluorescence values significantly increased compared to December, when the maximum of 1.14 mg m$^{-3}$ was recorded in the western shallow station ESAW-1 in the layer 40–50 m. A high value of 0.71 mg m$^{-3}$ was also observed in a thin surface layer of the distant station ESAW-GG in the same sampling period. DO concentrations were maximal at the surface, and lower between 400 and 800 m and between 800 m and the bottom (Figure 2l,m). Low DO values, together with high salinity values, are usually associated with the presence of LIW in the intermediate layer of the study area. Relatively low DO values in the deep layers, instead, indicate that there was not recent ventilation by dense water from the northern Adriatic shelf. However, along the western flank (at station ESAW-03) the presence of a vein of fresh and cold water of north Adriatic origin was found, sinking down to about 400 m.

During both sampling periods salty and warm waters advected from the Ionian Sea in the upper layer along the eastern flank into the SA were not confined only to the narrow coastal area, but spread toward the centre of the SA, reaching at least the station ESAW-09. The intermediate and deep layers were relatively homogenized horizontally between the eastern and western flank.

Vertical density distribution along the SA transect (not shown) followed to a great extent those of temperature and salinity. Hence, observed upwelling of the subsurface and intermediate isopycnal layers, especially in April 2016 around station ESAW-07, pointed to a more developed cyclonic gyre activity, with possible rim meandering, and mesoscale eddies favouring the lateral exchange and mixing between the coastal and open sea waters.

Qualitatively, the vertical distribution of thermohaline properties along the shallow Middle Adriatic transect (Figure 1), where the maximum depth is about 170 m, is similar to that found in the upper 200 m of the South Adriatic transect, and therefore is not shown. This is evident especially along the western shore, influenced by fresh and cool coastal waters, as described in the following paragraph for the two sampling periods. In December a relatively cold (<15 °C) and fresh (S < 37.50) 20 m thick surface layer was extending about 20 km offshore from the western coastal area up to the station ESAW-18 (data not shown). This situation was probably due to a more effective autumnal cooling of the North Adriatic water carrying fresh water from the Po River outflow, as compared to the open sea and to the SA. Maximum fluorescence values characterized entirely the upper freshwater layer of the western coast, where also the dissolved oxygen values were maximal (up to 5.8 mL/L). This pattern extended to the station ESAW-18. Below the fresh upper layer at the western coast, there was a tongue of saline and warm water extending between 30 and 80 m depth ($\theta$ > 16 °C, 38.70 < S < 38.75), protruding offshore up to station ESAW-16. Below 80 m, the thermohaline properties were more uniform horizontally over the entire transect, gradually getting cooler and hence denser with increasing depth despite relatively low salinity values in the deep layer. In April 2016, the thermohaline properties revealed a much smaller horizontal thermal gradient between the two shores, with temperature ranging from about 14.4 to 15.6 °C, on the eastern side (not shown). The vertical thermal gradient however, persisted, distinguishing warmer upper layers from cooler deeper layers. The freshwater surface layer along the western coast was thinner (10 m) than in December (20 m). Fresh waters (S 35.6–37) had also low density (26.4–27 kg m$^{-3}$). DO concentration along the whole transect was higher in April 2016 than in December 2015, especially in the upper 20 m up to station ESAW-20. The pattern associated with the relative fluorescence maximum had a surface signature near the western coast, visible to station ESAW-18, and a subsurface signature (between 40–60 m depth) in the central part of the transect (stations ESAW-18, 20, 22).

### 3.2. Vertical and Horizontal Distribution of Copepod Abundance

Figure 3 shows the vertical and horizontal distribution of the total copepod numbers (adults and copepodite stages) separated by the main orders. Copepodite stages contributed on average 23% to the total copepod numbers. Their contribution was greatest in the upper 100 m (25%), and decreased with depth. The bulk of copepod standing stock in terms of abundance was found in the upper 100 m layer in both sampling periods, with a general decrease in depth.

In December the average abundance in the upper 100 m was 131 ind. m$^{-3}$ with an exceptionally high value (632 ind. m$^{-3}$) recorded at the westernmost coastal shallow station ESAW-01 in the 0–50 m layer (Figure 3a). The lowest abundance in the upper 100 m was at station ESAW-03 (29 ind. m$^{-3}$). Only at station ESAW-12 a higher abundance was in the subsurface layer (50–100 m). In the 100–200 m layer average copepod densities were 27 ind. m$^{-3}$ with the maximum of 38 ind. m$^{-3}$ found at station ESAW-12. Below 200 m their numbers were uniformly low and not exceeding 13 ind. m$^{-3}$.

In April, in general, abundances were higher everywhere, except at the coastal station ESAW-01, compared to December (Figure 3b). In particular, the average abundance in the 0–100 m layer was 259 ind. m$^{-3}$ with the bulk of the population in the upper 50 m. This was particularly obvious in the open sea area (station ESAW-GG), where the maximum of 473 ind. m$^{-3}$ was recorded. Only at station

ESAW-05 a higher density value (351 ind. m$^{-3}$) was found in the subsurface layer. Average values of the copepod densities in the layers 100–200 and 200–300 m showed a decreasing trend (75 and 54 ind. m$^{-3}$, respectively). Below 300 m somewhat higher values were recorded at station ESAW-09, where in the layer 300–400 m copepod abundance was 50 ind. m$^{-3}$.

Order Calanoida dominated in abundance with an average contribution of 69%. Its maximum contribution was in December in the surface at station ESAW-01 (91%). Cyclopoida were the second order in terms of abundance (average contribution of 25%). The only two samples where cyclopoids dominated in terms of numbers were collected at station ESAW-09 in December, in the layers 400–600 and 600–800 m (57 and 48%, respectively) due to the higher numbers of Oncaeidae. Order Mormonilloida, with only one representative—*Neomormonilla minor*, contributed on average 4%. The presence of this species was more significant in December at the open sea stations: The greatest contributions were found at station ESAW-9, 300–400 m layer (19%), and station ESAW-GG, 200–300 m layer (16%). The order Harpacticoida contributed on average 2%.

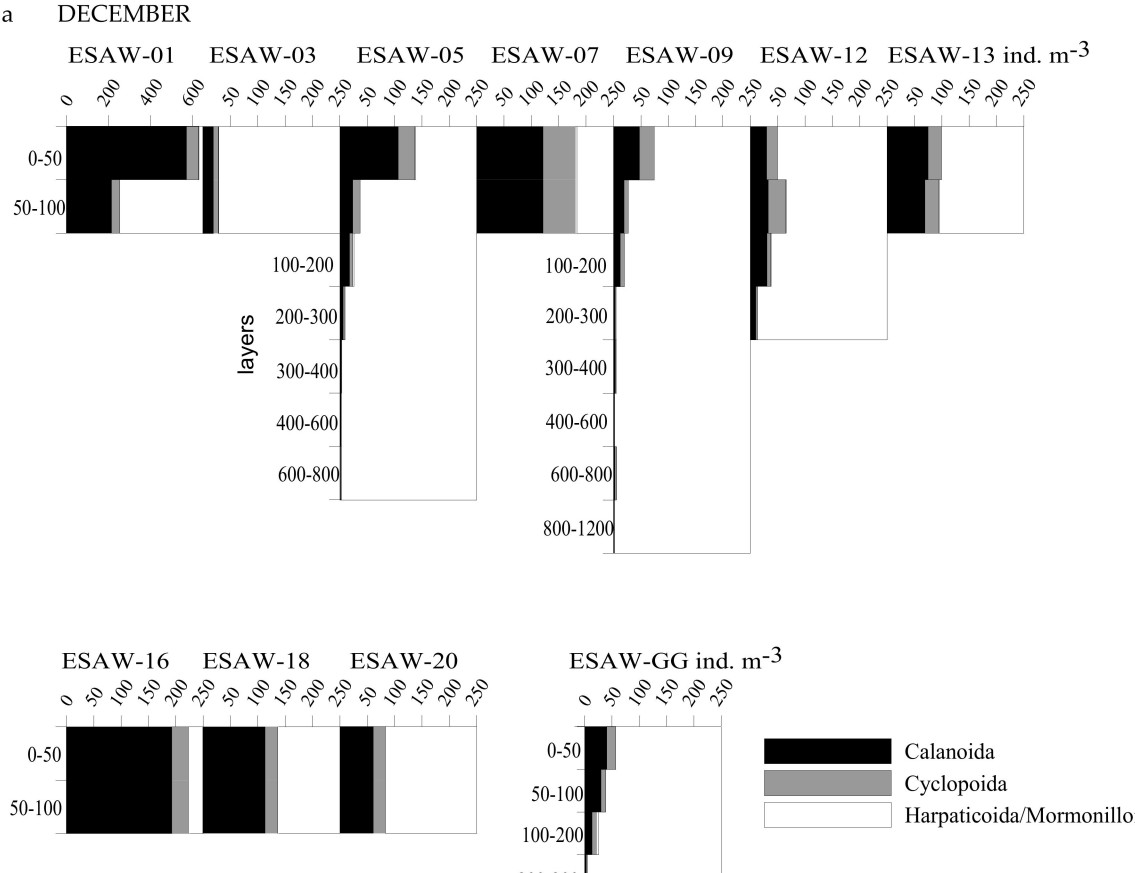

**Figure 3.** *Cont.*

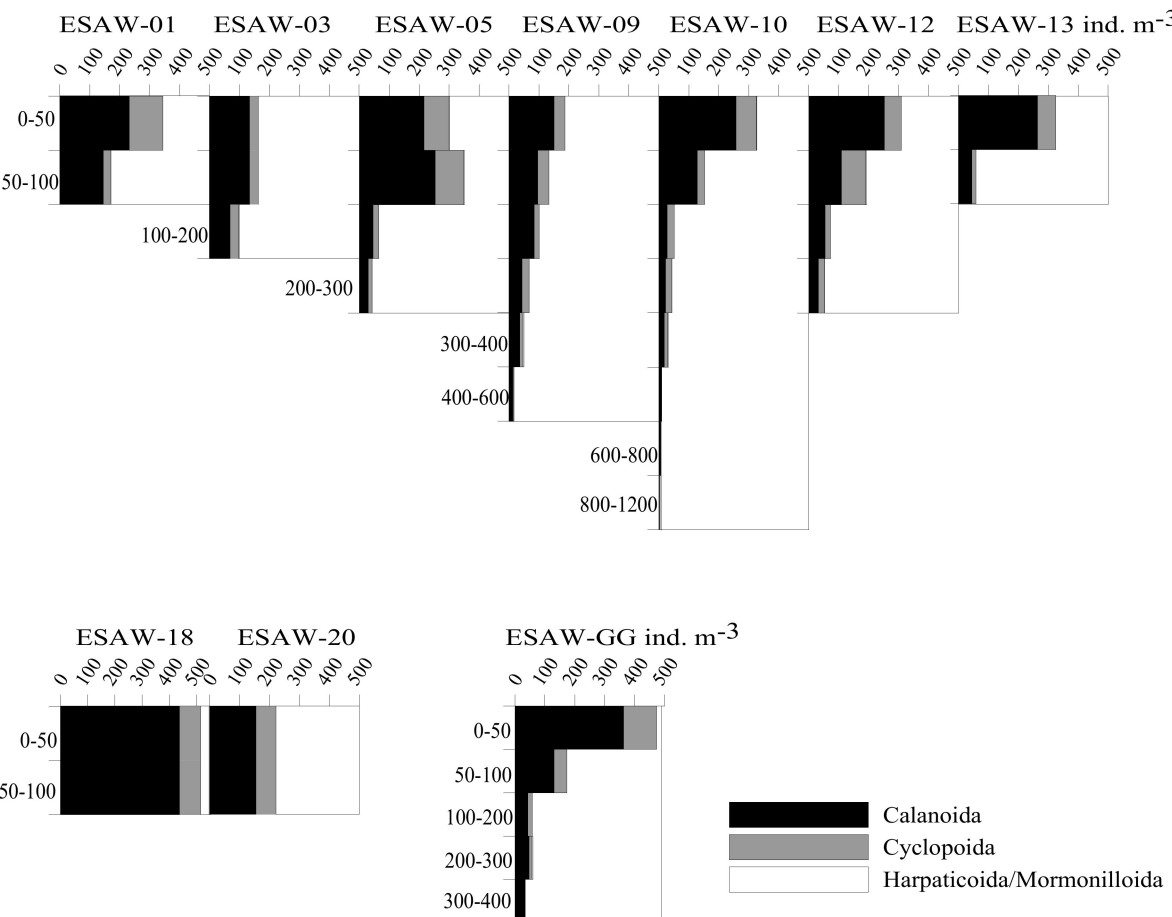

**Figure 3.** Spatial distribution of the total copepod densities in December 2015 (**a**) and April 2016 (**b**) along the investigated transect.

### 3.3. Copepod Composition and Diversity

In total, 110 taxa were found: 100 were recorded in December and 95 in April (Table S1). The most diverse group was order Calanoida with 85 species. Cyclopoida was presented with families Oithonidae (7 taxa), Oncaeidae, Lubbockiidaea (one species—*Lubbockia squillimana*), Corycaeidae and Sapphirinnidae (genera *Copilia*, *Sapphirina*, and *Vettoria*). Harpaticoida included three species (*Clytemnestra gracilis*, *Microsetella norvegica*, and *Macrosetella gracilis*), while order Mormonilloida included one species: *Neomormonilla minor.*

The average abundance of the taxa with a contribution larger than 1% was determined for each period (Table 2). The dominant copepod species was *Clausocalanus pergens*, followed by genus *Oithona* and *Haloptilus longicornis*. Abundant also were *Ctenocalanus vanus*, *Oithona plumifera*, and *Paracalanus parvus* whose densities, similar to most of the other taxa, were greater in April. By contrast, *Acartia* (*Acartiura*) *clausi* showed greater abundances in December due to the increased values at the western coastal stations (an exceptionally high value of 61 ind. m$^{-3}$ was found at station ESAW-01). The invasive calanoid *Pseudodiaptomus marinus* was found in December at the western coastal stations (ESAW-16, ESAW-18, and ESAW-01) in low numbers (<1 ind. m$^{-3}$).

**Table 2.** The most abundant copepod taxa (>1% contribution): Mean percentage in total numbers and mean abundance (ind. m$^{-3}$ ± SD) in December (a) and April (b).

| Taxa | Mean (%) | a | b |
|---|---|---|---|
| *Clausocalanus pergens* | 7.96 | 5.0 ± 8.4 | 10.6 ± 17.4 |
| *Oithona setigera*-group | 7.28 | 2.0 ± 2.4 | 3.8 ± 6.2 |
| *Oithona similis* | 6.79 | 4.8 ± 3.4 | 12.2 ± 13.1 |
| *Haloptilus longicornis* | 6.61 | 2.6 ± 4.0 | 3.4 ± 2.7 |
| *Oncaea* spp. | 6.24 | 1.4 ± 2.0 | 3.3 ± 3.8 |
| *Lucicutia flavicornis* | 5.52 | 2.1 ± 0.2 | 4.6 ± 6.9 |
| Corycaeidae | 5.28 | 2.2 ± 3.8 | 3.6 ± 3.4 |
| *Ctenocalanus vanus* | 5.27 | 3.6 ± 4.1 | 13.8 ± 23.8 |
| *Neomormonilla minor* | 4.0 | 0.9 ± 1.0 | 1.9 ± 0.9 |
| *Pleuromamma gracilis* | 2.75 | 0.8 ± 1.0 | 2.5 ± 3.7 |
| *Mecynocera clausi* | 2.66 | 2.5 ± 3.0 | 3.8 ± 4.0 |
| *Paracalanus parvus* | 2.31 | 4.6 ± 7.1 | 19.5 ± 28.9 |
| *Clausocalanus paululus* | 1.87 | 0.9 ± 1.4 | 2.8 ± 3.5 |
| *Clausocalanus jobei* | 1.63 | 4.2 ± 7.4 | 5.2 ± 7.3 |
| *Clausocalanus furcatus* | 1.57 | 1.8 ± 1.8 | 5.5 ± 5.7 |
| *Paraeuchaeta hebes* | 1.41 | 1.2 ± 1.5 | 5.9 ± 7.2 |
| *Monacilla typica* | 1.38 | 0.2 ± 0.1 | 0.4 ± 0.4 |
| *Calocalanus styliremis* | 1.35 | 2.0 ± 3.2 | 2.6 ± 1.9 |
| *Spinocalanus longicornis* | 1.32 | 0.2 ± 0.1 | 0.4 ± 0.4 |
| *Clausocalanus lividus* | 1.27 | 1.6 ± 2.5 | 3.2 ± 4.1 |
| *Acartia (Acartiura) clausi* | 1.17 | 10.8 ± 21.2 | 3.2 ± 5.1 |
| *Mesocalanus teniucornis* | 1.09 | 0.6 ± 0.7 | 1.6 ± 2.1 |
| *Pleuromamma abdominalis* | 1.08 | 0.2 ± 0.3 | 0.9 ± 1.3 |
| *Clausocalanus parapergens* | 1.05 | 0.4 ± 0.2 | 1.9 ± 1.3 |

Concerning the vertical species diversity, the greatest number of copepod taxa was recorded in December in the upper 100 m layer (Figure 4). In the same period an increased number of taxa was found in the 200–300 m layer. Apart from those two layers, a greater number of taxa were recorded in April compared to December with a marked decline in depth. In December, extremely high species richness (d) was found in the 800–1200 m layer, due to the low abundance of the copepods. In April, a maximum d was in the 600–800 m layer. The Shannon-Wiener diversity index (H′) ranged from 1.4 (below 800 m in December) to 2.7 (upper 100 m, also in December). In April, the maximum H′ of 2.6 was in 100–200 m depth strata and lowest diversity index values (1.8) were recorded in the layers below 800 m.

*3.4. Copepod Community Structure*

A hierarchical clustering distinguished five groups of samples at 45% similarity level (Figure 5). Their spatial distribution (Figure 6) revealed different patterns between December (Figure 6a) and April (Figure 6b), except for the first group on the western flank. Information on indicative copepod taxa of each cluster group is presented in Table 3.

The first cluster group included surface (0–50 m) and subsurface (50–100 m) samples from the western side of both sampling intervals, including also April surface samples from the eastern side. Within this group, the highest copepod densities were recorded. In total, 82 taxa were found; among these 26, according to their IV, were characteristic for this group. Generally, coastal copepod species were indicative for this group and many of them were found not only over the western flank but also in the surface layer of the open sea: *A. (Acartiura) clausi, Clausocalanus jobei, C. furcatus, Oithona similis* (ESAW-GG, ESAW-09, ESAW-10); *P. parvus, Temora stylifera* (ESAW-GG, ESAW-09), and *Centropages typicus* (ESAW-GG). Abundant taxa were also surface and subsurface species such as *C. vanus, Paraeuchaeta hebes,* and *C. pergens* as well as *Clausocalanus* juveniles.

The second cluster group was the most numerous (18 samples) and included the surface and subsurface samples collected in December from station ESAW-05 to the east. April samples belonging to this cluster group included subsurface samples and three samples from the layer below (100–200 m). This

group had the highest diversity index. In total, 82 taxa were found. The highest IV values were recorded for family Saphirrinidae, and species *Mecynocera clausi* and *Scolecithrix bradyi*. A high contribution of the juveniles and adults of the genus *Calocalanus* was found, as well as *Lucicutia flavicornis* and Corycaeidae. Within this group of samples, the same rare deep sea copepods were found (*Euchirella messinensis*, *Heterorhabdus abyssalis*, *H. spinifrons*). For example, *E. messinensis* was observed up to the surface layer in December (ESAW-05) and in the majority of the April samples.

The third cluster group concerns most of the mesopelagic samples collected in April and only two samples from December. Among 70 copepod taxa found in this group, *Chiridius poppei*, *N. minor,* and *Euchaeta acuta* showed the highest IV values. Abundant was also *H. longicornis*.

The fourth cluster group included the December mesopelagic layer, and only one April sample at station ESAW-10 (300–400 m). Compared to cluster 3, it has significantly lower densities within corresponding samples. In total, 65 taxa were recorded with only one indicative species, namely *Spinocalanus longicornis*. Except for the indicative species, higher average abundances were recorded for *H. longicornis* (3.4 ind. m$^{-3}$), *O. setigera*-group (2.8 ind. m$^{-3}$), and *N. minor* (1.2 ind. m$^{-3}$), while species that contributed the most were: *O. setigera*-group (19.5%), *N. minor* (15.4%), *Lucicutia flavicornis* (4.1%), and *Pleuromamma gracilis* (3.6%).

The fifth cluster group included December samples below 400 m and April samples below 600 m, comprising seven samples. Low abundances and in a total of 42 taxa were found, with IV >25 recorded for the deep sea copepods: *Temoropia mayumbaensis*, *Monacilla typica*, *Spinocalanus oligospinosus,* and *Candacia elongata*. Copepods that also contributed significantly were the family Oncaeidae (34.6%), *N. minor* (13.1%), and *S. longicornis* (6.1%).

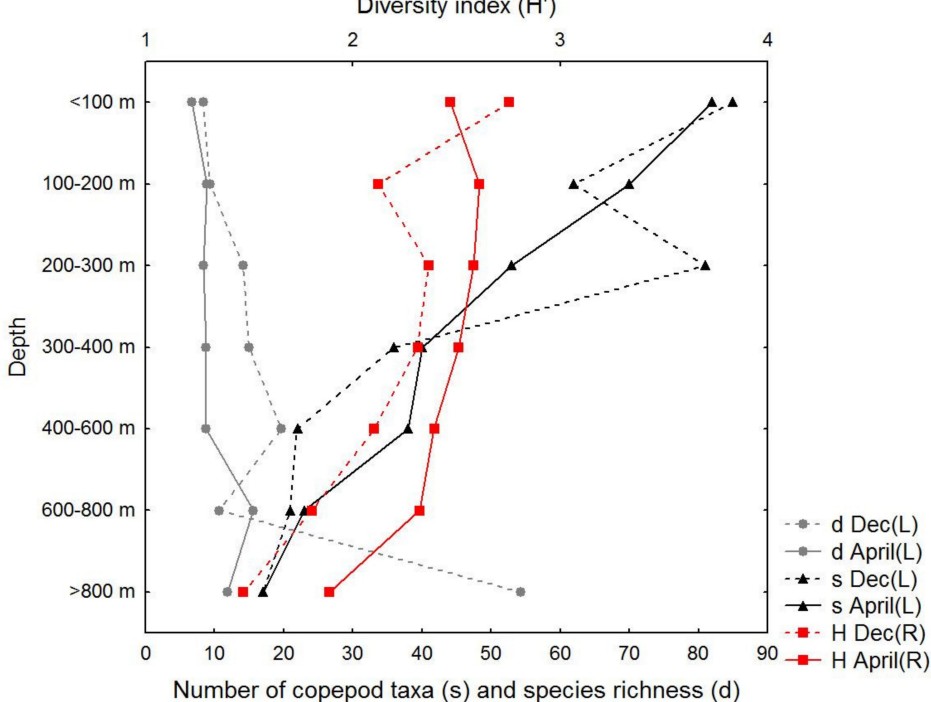

**Figure 4.** Vertical distribution of copepod diversity per sampling layer in December (dotted line) and April (solid line). S-number of taxa; H'-Shannon-Wiener diversity index, d-Margalef's species richness. L: *x*-axis bottom; R: *x*-axis top.

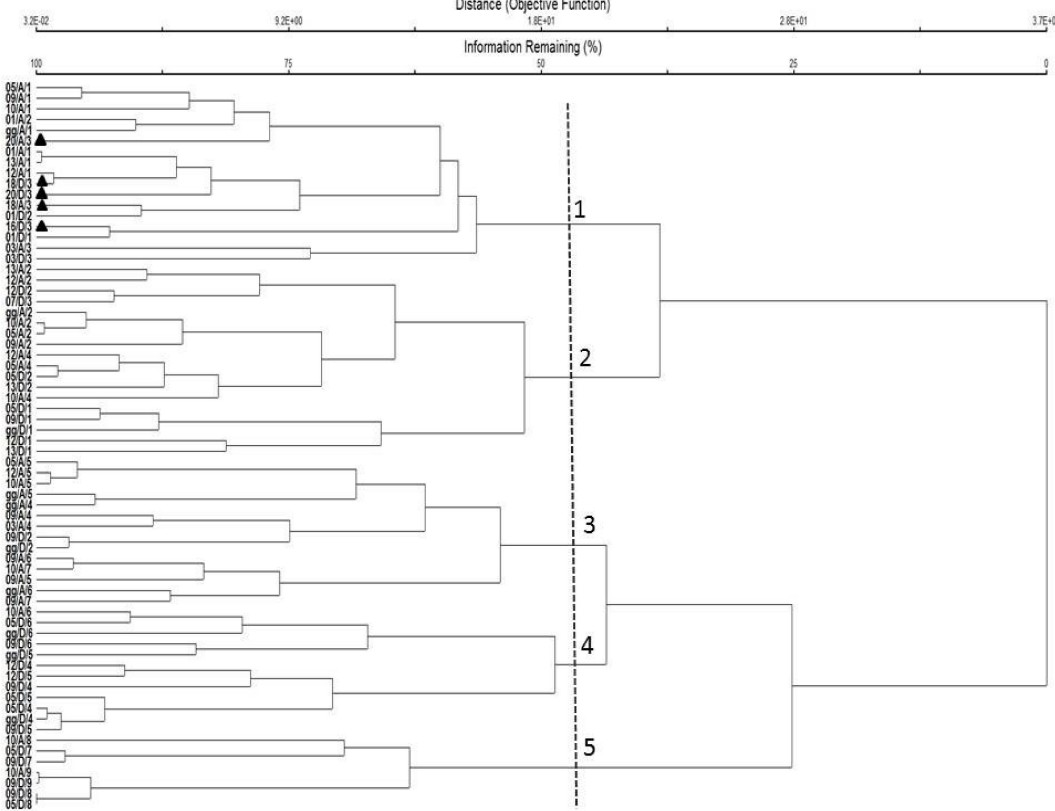

**Figure 5.** Cluster dendrogram of zooplankton samples taken in the Adriatic Sea in December 2015 and April 2016. Ordination is obtained by the relative Euclidean distance measure, using Ward's linkage method on the log-transformed abundance of the copepod taxa. Samples are indicated by the sampling station, sampling period (D = December, a = April), and sampling layer (1 = 0–50 m; 2 = 50–100 m; 3 = 0–100 m; 4 = 100–200 m; 5 = 200–300 m; 6 = 300–400 m; 7 = 400–600 m; 8 = 600–800 m; 9 = 800–1200 m). Samples taken on the western Mid Adriatic stations (ESAW-16, ESAW-18, and ESAW-20) are marked with a black triangle.

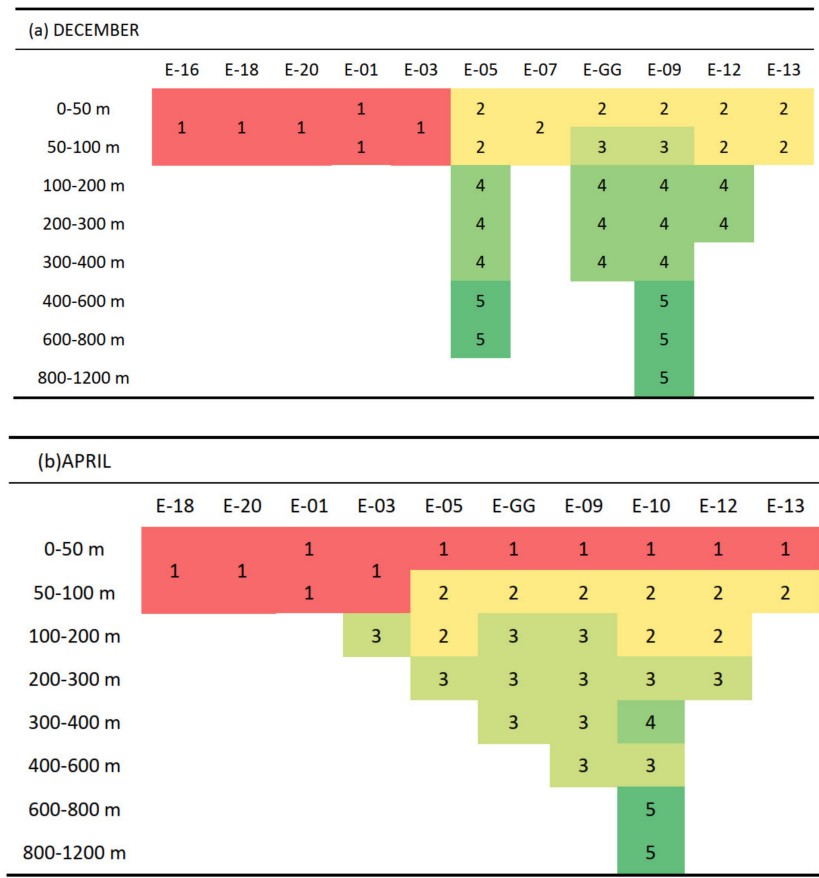

**Figure 6.** Vertical and horizontal distribution of the cluster groups over the investigated SA transect in December (**a**) and April (**b**); eastern coastal station: E-13, eastern midshelf station: E-12, eastern south central stations: E-09 and E-10, western south central stations: E-GG, E-07, and E-05, western midshelf stations: E-03 and E-20, and western coastal stations: E-01, E-16, and E-18.

**Table 3.** Copepod taxa that characterize the five clusters identified by the hierarchical clustering with the indicator values (IV), average abundance (Ab-ind. m$^{-3}$), and average contribution (C-%). Information on each cluster (average abundance, species richness, and diversity) are added.

| Cluster 1 | IV | Ab | C | | |
|---|---|---|---|---|---|
| *Acartia (Acartiura) clausi* | 85.1 | 6.8 | 3.6 | Av. Abundance | 264 ± 99 ind. m$^{-3}$ |
| *Paracalanus parvus* | 76.7 | 15.4 | 7.8 | d | 7.19 |
| *Clausocalanus furcatus* | 70.9 | 6.0 | 4.2 | H′ | 2.62 |
| *Paraeuchaeta hebes* | 67.4 | 6.3 | 3.7 | | |
| *Ctenocalanus vanus* | 62.3 | 24.3 | 13.6 | | |
| *Candacia giesbrechti* | 58.9 | 0.9 | 0.6 | | |
| *Calanus helgolandicus* | 57.2 | 1.1 | 0.8 | | |
| *Clausocalanus arcuicornis* | 57.0 | 2.7 | 1.4 | | |
| *Oithona similis* | 50.5 | 14.9 | 8.2 | | |
| *Centropages typicus* | 46.6 | 1.5 | 0.7 | | |
| *Diaixis pygmaea* | 45.3 | 0.57 | 0.5 | | |
| *Clausocalanus jobei* | 45.5 | 6.9 | 3.5 | | |
| *Clausocalanus pergens* | 44.5 | 19.8 | 11.5 | | |
| *Oithona plumifera* | 43.7 | 12.3 | 6.8 | | |
| *Neocalanus gracilis* | 43.0 | 0.6 | 0.5 | | |
| *Pareucalanus attenuatus* | 42.2 | 0.1 | 0.1 | | |
| *Temora stylifera* | 40.6 | 1.3 | 0.7 | | |
| *Clausocalanus lividus* | 40.5 | 3.3 | 3.1 | | |
| *Paracalanus nanus* | 39.1 | 1.1 | 0.6 | | |
| Oncaeidae | 36.0 | 4.3 | 2.9 | | |
| *Clausocalanus mastigophorus* | 35.8 | 2.0 | 1.1 | | |
| *Mesocalanus tenuicornis* | 34.7 | 1.0 | 0.7 | | |
| *Oithona nana* | 30.5 | 0.6 | 0.4 | | |
| *Calocalanus pavo* | 30.2 | 0.8 | 0.8 | | |
| *Nannocalanus minor* | 29.2 | 0.6 | 0.3 | | |
| *Clausocalanus* juv. | 25.7 | 5.4 | 2.7 | | |

**Table 3.** *Cont.*

| Cluster 1 | IV | Ab | C | | |
|---|---|---|---|---|---|
| Cluster 2 | | | | | |
| *Sapphirina* spp. | 76.8 | 0.3 | 0.4 | Av. Abundance | $111 \pm 49$ ind. m$^{-3}$ |
| *Mecynocera clausi* | 50.3 | 4.7 | 6.9 | d | 8.35 |
| *Scolecitrix bradyi* | 47.2 | 0.2 | 0.4 | H′ | 2.75 |
| *Calocalanus styliremis* | 46.9 | 2.5 | 3.0 | | |
| *Calocalanus contractus* | 46.1 | 0.9 | 1.5 | | |
| *Lucicutia flavicornis* | 41.9 | 7.0 | 8.5 | | |
| *Aetidaeus giesbrechti* | 41.9 | 0.2 | 0.3 | | |
| Corycaeidae | 40.5 | 4.8 | 7.1 | | |
| *Aetidaeus armatus* | 39.7 | 0.3 | 0.4 | | |
| *Scolecithricella dentata* | 36.0 | 0.5 | 0.6 | | |
| *Pleuromamma abdominalis* | 33.6 | 0.8 | 0.9 | | |
| *Calocalanus* juv. | 25.4 | 2.8 | 3.7 | | |
| Cluster 3 | | | | | |
| *Chiridius poppei* | 48.3 | 0.1 | 0.4 | Av. Abundance | $50 \pm 14.9$ ind. m$^{-3}$ |
| *Neomormonilla minor* | 38.0 | 1.7 | 5.7 | d | 8.93 |
| *Euchaeta acuta* | 37.3 | 0.3 | 1.1 | H′ | 2.54 |
| *Oithona setigera*-group | 35.0 | 4.6 | 14.6 | | |
| *Heterorhabdus spinifrons* | 34.7 | 0.2 | 0.4 | | |
| *Haloptilus longicornis* | 34.5 | 3.7 | 11.0 | | |
| *Clausocalanus parapergens* | 33.2 | 1.1 | 4.1 | | |
| *Pleuromamma gracilis* | 33.5 | 2.0 | 6.4 | | |
| *Lucicutia clausi* | 25.1 | 0.4 | 1.0 | | |
| Cluster 4 | | | | | |
| *Spinocalanus longicornis* | 27.5 | 0.2 | 3.2 | Av. Abundance | $16 \pm 9$ ind. m$^{-3}$ |
| | | | | d | 12.16 |
| | | | | H′ | 2.27 |
| Cluster 5 | | | | | |
| *Temoropia mayumbaensis* | 67.5 | 0.2 | 5.7 | Av. Abundance | $4 \pm 1.2$ ind. m$^{-3}$ |
| *Monacilla typica* | 44.0 | 0.3 | 14.0 | d | 22.1 |
| *Spinocalanus oligospinosus* | 43.9 | 0.1 | 2.9 | H′ | 1.96 |
| *Candacia elongata* | 28.6 | 0.004 | 0.2 | | |

A 2D NMS ordination solution (Figure 7) was chosen based on a stress value of 9.8. The final instability was <0.000001. Axis 1 describes the greatest amount of variation (89%) in the system, while Axis 2 described 5.7%. The first axis correlates positively with oxygen (Pearson's r = 0.904) and Chl-*a* (r = 0.807) values, and negative with layer (r = −0.843). Axis 2 is positively correlated with temperature (r = 0.421). The distribution of the samples (cluster groups) along the first two axes indicates the vertical distribution and is the principal factor that determines the variation in copepod assemblages. Species associated with the surface layers which are related to high DO values and Chl-*a* include *C. pergens, C. vanus, O. similis,* and *O. plumifera.* The oceanic *S. longicornis, M. typica,* and *N. minor* are associated with SA deep waters and low DO and Chl-*a* values. Species that showed negative correlations related to the second axis are *P. gracilis, P. abdominalis, L. flavicornis,* and *N. minor.*

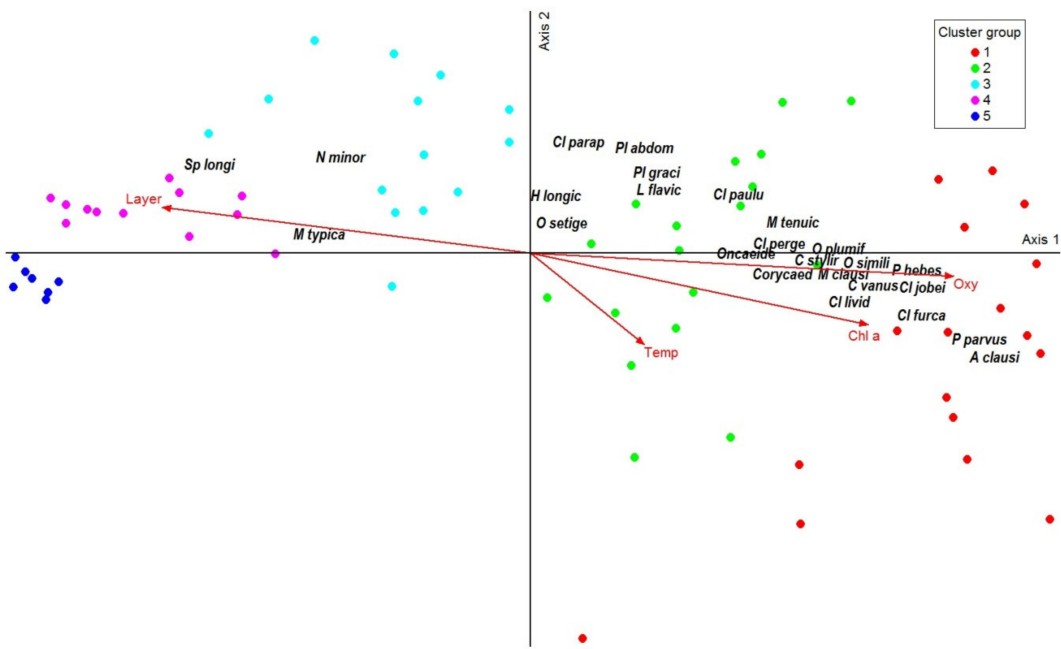

**Figure 7.** Ordination joint plot from the non-metric multidimensional scaling (NMS) with sample units labelled by cluster and position of the most abundant taxa with the environmental variables overlaid as vectors (Temp—Temperature, Chl-*a*, Oxy—DO, and Depth layer). Vector length and direction indicates relative strength of the correlation with axes. Abbreviation of analyzed copepod taxa: Cl perge—*Clausocalanus pergens*; O setige—*Oithona setigera*-group; O simili—*Oithona similis*; H longic—*Haloptilus longicornis*; L flavic—*Lucicutia flavicornis*; Corycaeid–Corycaeidae; C vanus—*Ctenocalanus vanus*; N minor—*Neomormonilla minor*; Pl graci—*Pleuromamma gracilis*; M clausi–*Mecinocera clausi*; P parvus—*Paracalanus parvus*; Cl paulu—*Clausocalanus paululus*; Cl jobei—*Clausocalanus jobei*; Cl furca—*Clausocalanus furcatus*; P hebes—*Paraeuchaeta hebes*; M typica—*Monacilla typica*; C stylir—*Calocalanus styliremis*; Sp long—*Spinocalanus longicornis*; Cl livid—*Clausocalanus lividus*; A clausi—*Acartia (Acartiura) clausi*; M tenuic—*Mesocalanus tenuicornis*; Pl abdom—*Pleuromamma abdominalis*; Cl parap—*Clausocalanus parapergens*.

## 4. Discussion

### 4.1. Copepod Abundance, Species Composition, and Diversity

As in the other regions of the Mediterranean Sea, copepods are the major component of the overall plankton in the SA and the general pattern followed by the mesozooplankton standing stock seems to be essentially driven by the dynamics of the copepod community. In open waters off Albania (SA) during May 2009 they contributed up to 90% of the mesozooplankton [46]. However, their share in total mesozooplankton can be reduced as in winter 2015 in the surface layer when they contributed to 67% of the total mesozooplankton due to the high proportion of appendicularians [19]. We have to note that the use of a large mesh size (250 μm) in the plankton net has omitted a very relevant component of the zooplankton community: Copepod nauplii and copepodites as well as adults of small species whose contribution in abundance and biovolume can be significant in the total community [56,57] and especially in the oligotrophic seas [26,58]. On the other hand, the mesh size used allows the comparison with a historical data and better insight into temporal changes in copepod abundance and composition.

In the Mediterranean Sea including the Adriatic Sea, the annual cycle of copepod densities usually peaks in the spring [32,45,59]. This is in accordance with our results, when in April 2016, copepod densities increased compared to the December 2015 survey. Furthermore, our total numbers found in both periods are in accordance with previous records during spring and autumn in the SA [19,44,46].

In general, the bulk of copepod abundance in the SA as well as in other Mediterranean regions [32,60–64] was found in the upper 100 m, and decreased with depth. Low copepod abundances at the central stations of the SA, as well as higher abundances at the perimeter stations towards the shorelines observed in December 2015 are a typical feature of the Adriatic Sea [44,45].

On the contrary, in April 2016, the horizontal abundance distribution in the upper layers was similar at coastal and open sea stations, which is unusual since coastal regions in the SA typically have higher densities than the open sea throughout the year [45]. We can hypothesize that the winter mixing, which reached approximately 400 m depth in winter 2016 [55] was responsible for nutrient enrichment of the euphotic zone and the consequent phytoplankton development, which in turn created conditions for the increased copepod abundance. A similar distribution of the total abundance of tintinnids was found over the same period [20] with the main bulk of the population in the subsurface layer at most of the stations, corresponding to a higher concentration of Chl-*a*. For the copepods, this was the case only at the western deep station ESAW-05, suggesting their higher grazing rate near the surface.

The copepod taxa identified during the ESAW cruises were in general agreement with the published data on the SA community [19,44,45]. Historical data of the four seasonal cruises in the SA recorded the most important species which constituted 26.4% of the total number of copepods: *C. pergens*, *C. paululus*, *P. gracilis,* and *L. flavicornis* [45]. Species that were quantitatively important and present in the Adriatic Sea in more or less constant numbers and are considered ubiquitous were: *C. vanus*, *O. plumifera*, *C. arcuicornis*, *C. rostrata,* and *C. helgolandicus*. In our study, a higher proportion of the genus *Oithona* was found, taken by the same mesh size, probably due to the seasonality of the *O. similis* and *O. setigera* whose annual peaks are in spring and winter, respectively. Domination of small size calanoids (*Clausocalanus*, *Paracalanus*, *Calocalanus*) and cyclopoids (*Oithona*, oncaeids, corycaeids) in epipelagic layers has already been observed in the Mediterranean [22,23,25,26,57,65–69]. Another copepod with relatively high contribution in our samples, *Haloptilus longicornis*, was present over the whole investigated area with greater densities than previously reported in the SA [44]. This species is common in the mesopelagic zone of the Adriatic Sea [44] as well as of the Ionian and Levantine seas [22,25,70] and is well adapted to the highly oligotrophic environment of the eastern Mediterranean Sea [71]. The presence of *H. longicornis* was probably favoured by the cyclonic phase of the BiOS mechanism and associated strong LIW ingression. On the contrary, during the anticyclonic phase of BiOS associated with a strong ingression of Atlantic water into Adriatic Sea especially in the early 1990s [17,72,73] *H. longicornis* was absent or registered in very low abundance (<1 ind. m$^{-3}$) [74].

Finally, the finding of the non-indigenous species *Pseudodiaptomus marinus* at the three stations off the western coast should certainly be emphasized. This copepod is typical for estuarine environments in the Northwest Pacific, whose spreading is mainly due to human shipping activities. The first record of *P. marinus* in the Adriatic was from the samples taken in 2007 near Rimini on the western coast [75]. Now, it seems to be widely spread through the Adriatic ports [76] and coastal areas (personal observation).

The number of copepod species found in various parts of the Adriatic Sea is more or less constant throughout the year, except in the SA, where it decreases in winter [45]. In total, we found more taxa in December, but the vertical distribution showed that, except for the surface and 200–300 m layer, more copepod taxa were recorded in April. The increased number of taxa in those layers in December could be due to inflow of Ionian Surface Water and Levantine Intermediate Water (LIW) into the Adriatic. The volume of this flow is greater in late autumn and winter [3,5]. A general decline in depth was obvious for a number of taxa found and for diversity values (H'). Observed values of species richness (d) showed an opposite trend. This evaluation of biodiversity is directly connected to the densities whose low values in the bottom layer in December caused an extreme species richness value. Compared to other records of the copepod diversity (H') in the Mediterranean Sea [22,32,77–79], our maximum values were significantly lower, which is probably because some families were not identified to the lower taxonomical levels.

### 4.2. Pattern of Copepod Assemblages in Relation to Environmental Parameters

A significant influence of winter vertical convection, lateral exchanges between coastal and open sea waters, and ingression of water masses of different properties occurred in between the two cruises. The structure of copepod communities, identified by the hierarchical clustering (at 45% similarity level), revealed five groups whose appearance differed between the investigated periods, in both vertical and horizontal distribution. The positioning of samples in the NMS plot shows a clear differentiation of cluster groups according to depth in the first axis which is in accordance with previous findings, where depth was considered as the primary habitat dimension for the copepod species [22,25,80,81]. In general, our observed vertical differentiation is in accordance with the depth zonation proposed by Scotto di Carlo et al. [32] with the epipelagic not extending deeper than 100 m, and the mesopelagic to 500–600 m, although the depth range of our copepod communities differs between the investigated seasons.

The first two clusters represent the epipelagic community, which varies in abundance, indicative species, and spatial distribution over the SA. In December, on a vertical basis the epipelagic layer (0–100 m) showed a differentiation found only between western near shore stations and central and eastern stations, in line with the distribution of water masses in the surface layer reflecting a massive presence of Ionian waters along the eastern margin of the SA and not the western one. At the central and eastern stations this layer contains species of the second cluster group mainly belonging to subsurface or even mesopelagic species, which match with the shallow region of the entrance of LIW and Ionian Surface waters. By contrast, in April the separation between the surface layer (0–50 m) and subsurface layer (50–100 m) in the open SA as well as on the eastern coast of the SA was evident. The well oxygenated surface layer and high Chl-*a* concentrations over the western coast, as well as its increased concentrations in April over the entire transect favoured the proliferation of copepod standing stock and the development of the copepod species attributed to the first cluster group. The vast majority of the copepods attributed to the first cluster were the members of coastal communities (*A. clausi*, *P. parvus*, *C. furcatus*, *C. jobei*, *O. similis*, *T. stylifera*, *C. typicus*) which spread over the open sea probably by lateral exchanges between coastal and open sea waters. Among the most numerous taxa, especially in April samples, were also two common Adriatic copepod species: *C. vanus* and *C. pergens* whose abundance peaks occur in spring [44]. A widening of the coastal species distribution from north to south and an increase in copepod abundance in the open waters of the Mid and South Adriatic was recorded also by Hure et al. [45]. This is particularly evident in spring and summer, due to the increment of Po river runoff, to the spreading of the fresh surface layer offshore [82], and due to the south-eastward surface currents. Consequently, mean density values and the population structure of Mid Adriatic western coastal waters are similar to those of the North Adriatic and markedly different from those of the east coast.

The second cluster revealed less than half lower average density values with respect to the first one. It represents the winter surface community of the central and eastern SA, and subsurface/upper mesopelagic community of the same area in April. This group of samples is also attributed to higher temperatures than the first one. The indicative species are mainly subsurface species, although intermediate *P. abdominalis* and *S. dentata* were also found as indicative. Along the eastern shore, there was an enhanced inflow of salty surface Ionian and Levantine waters from the adjacent Ionian Sea both in 2015 and 2016, amplified by the cyclonic mode of the North Ionian Gyre [83]. The volume of this flow is usually greater in late autumn and winter [3,5]. This is in accordance with higher diversity among this group of samples and quantitatively important taxa found in cluster 2 (Corycaeidae, *Lucicutia flavicornis*, *Mecynocera clausi*, *Calocalanus*) that are more abundant or present mainly in the eastern Mediterranean basin [23,67] than in the western one [64]. The influence of the waters originating from the eastern Mediterranean or the Ionian Sea was evident also by the tintinnid species composition of the surface layers of the eastern coast in December [20]. Considering the copepod community, it appears that the spreading of those waters was not limited to the eastern coast but extended to the entire central part of the SA, as thermohaline properties indeed showed. Additionally, the population shift of some of the

most abundant subsurface species toward the surface layer in December 2015 was recorded also for the tintinnids [20]. This is probably a consequence of the interaction between different waters (saltier waters of the Ionian origin along the eastern shore and fresher waters along the western side) able to spread toward the open sea within the SA cyclonic gyre driven by the intense mesoscale dynamics that characterizes the region.

The mesopelagic community of the SA was clustered in the third and fourth groups, which showed the lowest separation in the hierarchical clustering. Those groups were characterized by deep and subsurface copepod communities, with high abundance of *Haloptilus longicornis*. The variations were the most evident in copepod abundance, as well as in vertical distribution: In April this copepod community was found deeper than in December. The intermediate zone extended from 100 to 600 m [32] but upper and lower limits of this zone are difficult to define due to a gradual transition from one boundary layer to another and the constant diel shifting of intermediate-water species. Most of the samples taken in April in this zone were the part of Cluster 3 and included also two subsurface layers of SA open sea stations in December. According to IV this group was characterized by nine subsurface and deep open sea species among which are some strong migrants (*E. acuta* and *P. gracilis*) [32,64,84].

In December the mesopelagic community was more or less uniform within the SA and was distributed between 100 and 400 m (Cluster 4). The mesopelagic assemblage in December had only one characteristic species: *S. longicornis*. This species is quantitatively one of the most important deep sea species in the SA [44] and was also recorded as relatively abundant in the mesopelagic zone and deep waters of the Tyrrhenian, Ionian and Levantine seas [22,32,71,85]. In the Aegean Sea *Spinocalanus* spp. was found to be more abundant in early autumn compared to spring, suggesting adaptation of the genus to the highly oligotrophic waters [63]. Earlier, in late winter 2015 [19], some typical surface species (*P. parvus, A. clausi, C. typicus, T. stylifera*) were found in mesopelagic and deep layers (max. 100–400 m) in the SA (personal observation). This could be a consequence of two specific hydrological events: Winter vertical mixing [14,83] and occurrence of unusual feature with double salinity maximum layers (~50–200 m and ~400–600 m) with a relative salinity minimum layer in between them registered in 2015 [55]. These less salty waters that likely formed in the northern basin [86] were denser than the surface waters and balanced in depth between 200 and 500 m and advected in the centre of the SAP since the beginning of 2015 [55]. These specific features were registered in the SAP also in December 2015. However, during this cruise, there was no evidence of coastal and estuarine species in the mesoplegic layer as expected because these copepods can survive in this environment for a short time. In addition, recent research by the Acoustic Doppler Current Profiler (ADCP) in the open waters of SA has shown that crustacean zooplankton (mostly copepods) transported to deeper waters by vertical convection return to its preferred depth strata within 3–4 days [84]. In distinction to copepods, protozoans of relatively low motility are more useful indicators of horizontal and vertical transport [20,87–90].

The Mediterranean waters below 600 m are characterized by the absence of deep-sea species and their consequent replacement by several midwater species adapted to living at great depths [32]. Scotto di Carlo et al. [85] pointed out the uniformity of copepod species composition in the lower mesopelagic and the bathypelagic layers at basin scale. However, in some parts of the Mediterranean, for example in Aegean Sea, spring–early autumn changes in the dominant species composition were detected below 700 m [63] where the trophic conditions and hydrology have been found to affect community composition of the bathypelagic layer. Our result confirmed the uniformity of the deep sea communities, with the same copepod association (Cluster 5) was present in the deep layer during both sampling periods. Indeed, the temporal changes in community structure appear to affect only surface populations, showing little or no diel or seasonal variability in deep layers. However, most of the lower zone copepods in the deep Adriatic showed weak rise towards the surface in winter compared to spring [45]. This is even more evident for the mesopelagic species whose seasonal migrations are more obvious with a rise towards the surface in winter and abrupt descent at the beginning of spring. Thus, the vertical zonation of the cluster groups indicates that mesopelagic and deep copepod community appear at the lower depths in December than in April. Low abundance and typical deep

sea species were indicative of the fifth cluster group. Furthermore, we found a significant contribution of cyclopoids, which is due to the high densities of the Oncaeidae, especially in December in the deep layer. They have been found to peak in the mesopelagic zone of several areas such as the Red, Levantine and Arabian seas [91–93], the western subarctic Pacific Ocean [94], the Arctic Ocean [95], and their important contribution to mesozooplankton down to 1000 m has also been recorded for the Aegean Sea [63]. This family together with Corycaeidae feed on appendicularian houses and other types of marine snow aggregates [96–98] and shows a capacity for living under very oligotrophic conditions [99]. This indicates that sinking particles can be an important source for copepods inhabiting the deepest layers of SA, since most of them do not migrate or are weak or even inverse migrants [32,91].

## 5. Conclusions

During investigated periods (December 2015, April 2016) physical forces such as winter vertical convection, lateral exchanges between coastal and open sea waters, and ingression of LIW were able to regulate the spatial distribution of copepods in the SA. These processes have an impact on Chl-*a* and oxygen distribution which are, alongside with depth, the most important factors in the separation of the copepod assemblages in the SA. Two different types of copepod horizontal and vertical distribution can be defined for the two periods. In December, the coastal copepod community was limited only to the western flank. The epipelagic areas of the open SA as well as of the eastern coast were characterized by high diversity, low abundances in the central area and subsurface/upper mesopelagic copepod species. In April, abundances over the entire vertical profile were higher than in December. The copepod horizontal distribution showed higher abundances in the surface area of the central SA, which was probably a consequence of late-winter/early spring blooms near the centre of the SA. At that time, the surface coastal copepod community was present along the entire transect while subsurface and mesopelagic copepod communities were deeper than in December. Mesopelagic fauna of both months had high abundances of *Haloptilus longicornis*, characteristic species of the eastern Mediterranean, whose presence was probably favoured by the cyclonic phase of the BiOS mechanism and the consequent stronger LIW ingression into the SA. Further studies at a more detailed taxonomical level including smaller copepods are essential for a better estimation of copepod distribution related to winter physical forcing. In addition, it would be very interesting to study the entire copepods seasonal cycle in order to depict the degree of spatial and temporal variability in this highly dynamic and diversified environment, which could provide us with an idea of future changes in zooplankton dynamics and consequently its implications for fisheries.

**Supplementary Materials:** The following are available online at http://www.mdpi.com/2077-1312/8/8/567/s1, Table S1: The copepod taxa identified in the Adriatic Sea in December 2015 (D) and April 2016 (A). Stations are grouped by geographical position, as followed: ECS (eastern coastal station: ESAW-13), EMS (eastern midshelf station: ESAW-12), ESC (eastern south central stations: ESAW-9 and ESAW-10), WSC (western south central stations: ESAW-GG, ESAW-7 and ESAW-5), WMS (western midshelf stations: ESAW-3 and ESAW-20), WCS (western coastal stations: ESAW-1, ESAW-16 and ESAW-18). Roman numerals indicate depth layer (I = 0-50 m, II = 50-100 m, III = 100-400 m, IV = 400-1200 m).

**Author Contributions:** Zooplankton analysis and writing (original draft preparation), M.H.; interpretation of the data of the article and discussion of their results, M.H., M.B. (Mirna Batistić), V.K., and M.B. (Manuel Bensi); review and editing, M.B. (Mirna Batistić); Zooplankton sampling and paper editing, R.G.; physical oceanography data (contribution to results and discussion), V.K. and M.B. (Manuel Bensi). All authors have read and agreed to the published version of the manuscript.

**Funding:** This research was carried out within the framework of the EUROFLEETS2 research infrastructures project under the 7th Framework Programme of the European Commission (grant agreement 312762). The research was also supported by the Italian national project RITMARE (Ricerca Italiana per il Mare, grant numbers: SP3-WP3-AZ1; SP4-LI4-WP1; SP5-WP3-AZ3) and the Croatian Science Foundation, under project IP-2014-09-2945.

**Acknowledgments:** We thank the captain and the crew of R/V Bios dva and all the scientific and technical staff who took part in the cruises. The bottom morphology along the transect is composed partially from the echo-sounding data recorded during the campaign and elaborated by V. Tičina (Institute of oceanography and fisheries, Split, Croatia) and partially from the GEBCO 2014 1 × 1 min grid for the Mediterranean Sea. We also thank Jakica Njire for her help on graphical presentation of the results and to Steve Latham for English editing of the manuscript.

**Conflicts of Interest:** The authors declare no conflict of interest. The funders had no role in the design of the study; in the collection, analyses, or interpretation of data; in the writing of the manuscript, or in the decision to publish the results.

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
