# Peer review of "Copepod Community Structure in Pre- and Post- Winter Conditions in the Southern Adriatic Sea (NE Mediterranean)"

_jmse, doi:10.3390/jmse8080567_

Round 1
Reviewer 1 Report
The paper entitled "Copepod community structure in Pre-and Post-Winter Conditions in the Southern Adriatic Sea (NE Mediterranean)” is devoted to know the structure of the copepods community in two different months by depth stratified samples from the surface to 1200 m depth in several transects of the South Adriatic (SA). The authors show firstly the complex dynamic of the study area to explain finally the distribution of the zooplankton community in relation to the strong physical forces in the SA. They explain the two physical and biological situations in December and April, their similarities and differences and their copepod assemblages in both periods and depth preference. Although is described its distribution in the whole water column is particularly interesting the description of this community in the mesopelagic strata. They also found a higher abundance in the central Adriatic Sea besides the coastal one, probably due to the winter convection in spring, as they mention, which is very important for this oligotrophic sea.
After my lecture and review I can say that the paper is very interesting and well written, giving important information on the zooplankton community in the SA and in general, for the Eastern Mediterranean Sea. The copepod community was quite well described although their taxonomical level was not very detailed particularly the “non calanoids”. But they mention at the end of the study this particularity for further research in the area.
The abstract gives a synopsis of the paper very clear as well as all the different paragraphs well written and described. The introduction is well documented and the state of the art is good for the area although I would add some more interesting references. Methods and results are adequate to the description of the main conclusions. Finally the discussion is quite extended but clearly developed. Figures and tables are all necessary.
However there are several comments to clarify and from my point of view few corrections that have to be included to improve the manuscript.
After that minor revisions below indicated, for me the manuscript can be ready for publish.
Introduction:
There are references with years instead the number of the references which have to be removed. Line72, 75, 77, 80……
Materials and methods
The units of zooplankton always have to be similar or ind./m3 or ind. m-3 . Line 122 and others
Chlorophyll in line 93 no in capital letter
Shannon–Wiener diversity better than Shannon diversity only in different sentences, line 128 as in line 271
Please explain why it was used the Euclidean distance for hierarchical clustering for zooplankton assemblages.
Results
Please remove bibliography in this paragraph (20 and 54)
Correct units and write similarly, see line 217 and line 221 or line 222 etc….
Line 200 please explain better why is denser and low salinity. It is not clear to me.
Line 205 write again because it is not clear
Discussion
It is very clear and well documented but I found the lack of some bibliography of the oligotrophic western Mediterranean see Fernandez de Puelles et al, 2007, Andersen et al., 2001; 2004, Brugnano et al, 2012
And also the reference about the vertical distribution of copepods in the open and deep three main oceans (Fernandez de Puelles et al., 2019)
Remove Pit in line 383 and also with the same mesh size. Not necessary.
In line 504 add bibliography from the western side after western one
In line 524 add reference
In line 534 I cannot understand why in 600 m the waters are characterized by bathypelagic species because is not yet bathypelagic depth. According to the sampled depth I do not consider we can speak in any moment about bathypelagic species. Please explain better or change the phrase.
Remove also names of the authors and years by the numbers in the references such as line 435, 448, 477
In figure 4 to correct sapling by sampling
Figure 5 it is not clear
In figure 7, I cannot see the names of the species, it will be better with a bigger resolution
In supplementary material
Indicate meaning of A and D. I know that is April and December but has to be written
Author Response
REWIEVER#1:
Answer (A): We thank reviewer for useful suggestions and corrections, which we have all respected. The specific comments are listed below.
Introduction:
There are references with years instead the number of the references which have to be removed. Line72, 75, 77, 80……
A: the changes were made (lines 72, 74, 76, 79, 454).
Materials and methods
The units of zooplankton always have to be similar or ind./m3 or ind. m-3 .
Line 122 and others
A: the changes were made (lines 121, 237).
Chlorophyll in line 93 no in capital letter
A: the change was made (now line 92).
Shannon–Wiener diversity better than Shannon diversity only in different sentences, line 128 as in line 271
A: the change was made (now lines 127 and 132).
Please explain why it was used the Euclidean distance for hierarchical clustering for zooplankton assemblages.
A: we explained it (line 139).
Results
Please remove bibliography in this paragraph (20 and 54)
A: we removed it (lines 164, 169).
Correct units and write similarly, see line 217 and line 221 or line 222 etc….
A: it was corrected.
Line 200 please explain better why is denser and low salinity. It is not clear to me.
A: We have modified the sentence to make the concept clearer. In particular, this is the typical case when a temperature decrease has more influence on the density increase, not compensated by low salinity values. The new phrase is:
“Below 80 m, the thermohaline properties were more uniform horizontally over the entire transect, gradually getting cooler and hence denser with increasing depth, despite relatively low salinity values in the deep layer” (line 213).
Line 205 write again because it is not clear
A: The phrase “The fresh surface layer along the western coast was shallower (10 m) than in December.” was modified as follows:
“The freshwater surface layer along the western coast was thinner (10 m) than in December (20 m)” (line 219).
Discussion
It is very clear and well documented but I found the lack of some bibliography of the oligotrophic western Mediterranean see Fernandez de Puelles et al, 2007, Andersen et al., 2001; 2004, Brugnano et al, 2012
And also the reference about the vertical distribution of copepods in the open and deep three main oceans (Fernandez de Puelles et al., 2019)
A: the references were added (lines 396, 513).
Remove Pit in line 383 and also with the same mesh size. Not necessary.
A: it was corrected (lines 379-400).
In line 504 add bibliography from the western side after western one
A: it was added (line 493).
In line 524 add reference
A: it was added (line 525).
In line 534 I cannot understand why in 600 m the waters are characterized by bathypelagic species because is not yet bathypelagic depth. According to the sampled depth I do not consider we can speak in any moment about bathypelagic species. Please explain better or change the phrase.
A: we rearranged text according to the suggestion (line 535).
Remove also names of the authors and years by the numbers in the references such as line 435, 448, 477
A: we removed it (lines 446, 457, 478).
In figure 4 to correct sapling by sampling
A: it was corrected.
Figure 5 it is not clear
A: we added explanation in the figure capitation.
In figure 7, I cannot see the names of the species, it will be better with a bigger resolution
A: the change was made.
In supplementary material
Indicate meaning of A and D. I know that is April and December but has to be written
A: the change was made.
Reviewer 2 Report
The authors attempt to present a picture of copepod abundance, composition, and distribution in relation to hydrophysical conditions in the middle and southern Adriatic in pre- and post-winter periods (December 2015 and April 2016).
Since the copepods of the Adriatic Sea have been intensively studied in recent years, the findings on the vertical distribution of species and their abundance, as well as on species diversity, are not novel (see for example Hure et al. 1980; Hure and Krsinic, 1998, and others).
My main concern is the use of the net with coarse mesh (250 µm) to collect small sized zooplankton. The authors explain this choice by the fact that a similar mesh size was used earlier which “allows the comparison with a historical data and better insight into temporal changes in copepod abundance and composition”. (line 375). However, comparing the obtained data with historical data is not the announced goal of the work. But in the case of assessing the species richness and biodiversity, an underestimated number of copepods can distort the true values of these parameters.
Another important weakness of this work is the absence of a sound discussion on the relationship between copepod distributional patterns and patterns of hydrophysical events. For example, the influence of the lateral exchange between shelf and open sea on the copepod distribution across the transect is not supported by physical data obtained in the same cruise. The high abundance of Haloptilus longicornis is explained by strong intrusion of the Levantine Intermediate Waters. It would be rational to show that abundance of this species is reduced at low LIW intrusion. The authors hypothesize that the increase in the copepod number in the open sea is caused by winter deepwater convection. Was the alternative hypothesis tested?
This manuscript in its present form does not provide novel and relevant information about the distribution of the copepod assemblage in relation to hydrophysical pattern in the Adriatic Sea.
I suggest the resubmission of this paper after revision of the section “4.2. Pattern of copepod assemblages in relation to environmental parameters”
Specific comments:
Fig 3. Vertical – remove the dividing line between layers 0-50 and 50-100m when sampling was 0-100m
Fig 4 (Caption) change the R: y-axis top to the x-axis top
Line 415. The first mention of the species in paragraph should be done with full genus and species
Table S1 No definitions of layers. Instead of meters – I, II ets
Author Response
REWIEVER#2:
Answer (A): We thank the reviewer for valuable and constructive remarks. We have rearranged discussion and changes were made through the text. The specific comments are listed below.
Since the copepods of the Adriatic Sea have been intensively studied in recent years, the findings on the vertical distribution of species and their abundance, as well as on species diversity, are not novel (see for example Hure et al. 1980; Hure and Krsinic, 1998, and others).
A: In the papers of Hure et al., 1980 and Hure and Kršinić 1998, copepod data are taken on the cruises conducted in middle of 1970s. Given the global changes in recent decades and new insights into Adriatic Sea circulation, we consider that our results give valuable information on present copepod population structure of the Adriatic Sea.
My main concern is the use of the net with coarse mesh (250 µm) to collect small sized zooplankton. The authors explain this choice by the fact that a similar mesh size was used earlier which “allows the comparison with a historical data and better insight into temporal changes in copepod abundance and composition”. (line 375). However, comparing the obtained data with historical data is not the announced goal of the work. But in the case of assessing the species richness and biodiversity, an underestimated number of copepods can distort the true values of these parameters.
A: This is a general problem, because one mesh size cannot cover all of copepod and zooplankton size fractions in total. We emphasized this problem in the discussion and we are aware of the underestimation of smaller copepod taxa. On the other hand, since the South Adriatic is an oligotrophic system and most of the data were obtained in the open/deep sea with huge depth ranges, the coarser net with larger diameter allowed us to catch larger copepod species that contribute significantly in such areas. Furthermore, the study of the mesh size effects on mesozooplankton community in the South Adriatic (Miloslavić et al., 2014) showed little or no significant difference in species richness and biodiversity between samples collected with different mesh sizes.
Another important weakness of this work is the absence of a sound discussion on the relationship between copepod distributional patterns and patterns of hydrophysical events. For example, the influence of the lateral exchange between shelf and open sea on the copepod distribution across the transect is not supported by physical data obtained in the same cruise. The high abundance of Haloptilus longicornis is explained by strong intrusion of the Levantine Intermediate Waters. It would be rational to show that abundance of this species is reduced at low LIW intrusion. The authors hypothesize that the increase in the copepod number in the open sea is caused by winter deepwater convection. Was the alternative hypothesis tested?
A: we added additional information about reduced H. longicornis abundances during low LIW intrusion (lines 425-429). Also, some other changes were made through the discussion section in order to make more clear relationship between copepod distribution patterns and hydrographical events.
This manuscript in its present form does not provide novel and relevant information about the distribution of the copepod assemblage in relation to hydrophysical pattern in the Adriatic Sea.
I suggest the resubmission of this paper after revision of the section “4.2. Pattern of copepod assemblages in relation to environmental parameters”
A: we revised the entire discussion section according to the comments.
Specific comments:
Fig 3. Vertical – remove the dividing line between layers 0-50 and 50-100m when sampling was 0-100m
A: we removed it.
Fig 4 (Caption) change the R: y-axis top to the x-axis top
A: it was changed.
Line 415. The first mention of the species in paragraph should be done with full genus and species
A: it was changed (line 430).
Table S1 No definitions of layers. Instead of meters – I, II ets
A: the definition was added.
Round 2
Reviewer 2 Report
Accept in present form
Author Response
We thank reviewer for the useful comments and suggestions.